# The Growth of Metal–Organic Frameworks in the Presence of Graphene Oxide: A Mini Review

**DOI:** 10.3390/membranes12050501

**Published:** 2022-05-06

**Authors:** Nurul A. Mazlan, Fraz Saeed Butt, Allana Lewis, Yaohao Yang, Shuiqing Yang, Yi Huang

**Affiliations:** 1School of Engineering, Institute for Material & Processes, The University of Edinburgh, Robert Stevenson Road, Edinburgh EH9 3FB, UK; n.a.mazlan@sms.ed.ac.uk (N.A.M.); f.s.butt@sms.ed.ac.uk (F.S.B.); a.m.lewis-6@sms.ed.ac.uk (A.L.); 2Jiangsu Dingying New Materials Co., Ltd., Changzhou 213031, China; yaohao.dingyingmaterials@gmail.com (Y.Y.); dingyingmaterial@163.com (S.Y.)

**Keywords:** graphene oxide, metal–organic frameworks, MOF/GO composite, directed growth, orientation

## Abstract

Integrated metal–organic frameworks (MOFs) with graphene oxide (GO) have aroused huge interest in recent years due to their unique properties and excellent performance compared to MOFs or GO alone. While a lot of attention has been focused on the synthesis methodologies and the performance analysis of the composite materials in recent years, the fundamental formation/crystallization mechanism(s) is (are) still not fully understood. Ascribed to the distinctive structural and functional properties of GO, the nucleation and crystallization process of MOFs could be altered/promoted, forming MOF/GO composite materials with different nanostructures. Furthermore, the MOF’s parental structure could also influence how the GO and MOF bond together. Thus, this short review attempted to provide critical and indepth discussions of recent research results with a particular focus on the factors that influence the directional growth of parent MOFs in the presence of graphene oxide. Due to the unique structure and enhanced properties, the derived MOF/GO composites have a wide range of applications including gas separation, electrochemistry, and photocatalysis. We hope this review will be of interest to researchers working on MOF design, crystal structure control (e.g., orientation), and composite materials development.

## 1. Introduction

Metal–organic frameworks (MOFs) also known as porous coordination polymers are a class of advanced materials designed by employing various metal ions and organic linkers [1]. MOFs have attracted attention in both academia and industry due to interesting characteristics and unique features such as a high surface area with tunable porosity and physiochemical properties, the presence of unsaturated metallic sites [2], and the ability to incorporate specific active species without noticeable change or damage to the framework topology [3]. Due to these interesting features, the potential use of MOFs has been investigated in a wide range of applications such as photocatalysis [4], environmental remediation [5], water treatment [6], and energy storage [7].

Nevertheless, the practical use of MOF materials has been restricted from large-scale applications due to certain drawbacks including high fabrication and processing costs, poor chemical, thermal, and mechanical stability, low capacity, and difficulties in recycling/regeneration. Due to these properties, many MOFs are not effective for small molecule adsorption applications such as ammonia [8]. Instead, MOFs can be combined with other suitable structures, such as graphene-based materials, to improve the overall structural properties and expand their applications. Graphene oxide (GO) consists of layers where the planes of GO are decorated with oxygen functional groups [9,10,11,12]. Due to the coexistence of ionic groups such as the oxygen functional groups on the planes and aromatic sp^2^ domains [13], GO is involved in a wide range of bonding interactions with MOF structures.

To date, the synthesis of new MOF/GO composites has been studied extensively [14,15,16,17,18]. Many new types of composite materials were synthesized in the presence of GO with advanced properties and extraordinary performance. Ma et al. reported that a new type of ultrafiltration membrane incorporated with UiO−66/GO composite not only exhibited remarkable hydrophilic properties and high solute rejections but also demonstrated impressive antifouling performance [18]. However, the fundamental understanding of MOF growth and morphology control in the presence of GO has not been clear.

One of the critical aspects in the development of MOF/GO composite materials is to understand the influence of GO on the nucleation and growth of MOF structures. This fundamental knowledge could open up vast opportunities to tailor the nanostructure of MOF/GO composites with desirable properties and features, thus targeting many challenging applications. Our study on recent works indicates that the nucleation and growth of MOFs are greatly affected by the oxygen-containing functional groups on GO basal planes as well as the coordination of the metallic sites in parental MOF structures [8,15,19,20]. Furthermore, with GO participating in the crystallization process, MOF growth can be controlled in certain directions to form oriented MOF structures, favoring uniform crystal and pore structures, which are advantageous to many applications. Based on this review, the growth direction of MOFs could be simply achieved by manipulating the MOF fabrication strategies in the presence of GO via either bottom-up or top-down approaches [21,22,23,24]. Recently, MOF/GO composites with oriented MOF structures have gained increasing attention because of their uniquely oriented MOF structures and superior performance in gas separation/adsorption [13,21], electrochemical applications [25], and photocatalysis [26] as compared to the results reported for physical MOF/GO blends. Our review in this field suggests that various types of MOF/GO composite materials have been synthesized and evaluated [16,18,27,28,29], but a comprehensive understanding of the underlying factors that affect the MOF/GO composite formation, particularly the nucleation and growth of MOF structures have often been overlooked. Hence, as shown in Figure 1, this review summarized the possible factors which could influence the MOF growth in the presence of GO from many (but still limited) recently reported works. We also attempted to cover the major applications of such interesting MOF/GO composites, especially those with oriented MOF structures.

## 2. Factors Affecting MOF Growth on GO

As discussed above, preferential growth of numerous MOFs on GO basal planes has been observed. However, the understanding of MOF nucleation and growth on GO is still very limited because each MOF may follow a different nucleation–crystallization crystal growth pathway, and, as a result, they are extremely versatile in structures. Such structure versatility arises from the rich combination of abundant inorganic building units and organic linkers, which offers immense opportunities to form different MOF structures even in a simple solution-mediated synthesis system. The introduction of GO in MOF synthesis further complicates the synthesis system as well as the crystallization process. However, those contributing factors were grouped into three main categories, i.e., GO functional groups–metal binding site interactions, MOF coordination chemistry effect, and pre-formed GO guide.

GO nanosheets are abundant in oxygen species on their edges and defect sites. Multiple studies have been reported that highlight the importance of the interaction between oxygen-containing groups (e.g., –OH and =O functional groups) and MOF metal-binding sites in directing MOF nucleation and subsequent crystal growth [21,30,31]. Interestingly, it has also been found that functionalized GO with different oxygen functional groups could potentially gain the control of MOF growth direction on GO basal planes [31]. Secondly, the metal–ligand coordination, as well as the metal center chemistry of the participating MOF, can also affect MOF crystallization and eventually its growth pattern on the GO basal plane. This is because different types of MOFs have a different metal center and metal–ligand coordination, given the fact that the metal centers might have completely different bonding capabilities. Therefore, the interactions between metal centers and guest GO layers can also affect MOF nucleation and growth, causing the resulting MOF/GO structure to be ordered or disordered. On the other hand, the MOF geometry, including morphology, metal center configuration, and pore shape and arrangement, can be crucial in determining the resulting MOF/GO composite structure and, therefore, their chemicophysical properties [8,19,20]. Lastly, a few studies have also reported the possibility of guiding MOF growth by combining a bottom-up fabrication method and the existing oriented GO layer/coating [21,22,23]. The preformed GO layer plays a critical role in the formation of a highly oriented MOF structure. This is because the confinement of GO layers is capable of guiding the directional growth of the MOFs underneath, avoiding other random growth and thus promoting the formation of a highly oriented MOF structure. Furthermore, the GO layer/coating can act as structural nodes and participate in bonding interactions with MOFs. The aforementioned contributing factors are reviewed in detail in this work and discussed separately in the sections below.

### 2.1. Interactions between GO Functional Groups and MOF Metal-Binding Sites

Graphene oxide nanosheets are decorated with an abundance of oxygen functional groups in the form of epoxy with hydroxyl groups on the basal planes, and carboxylic and sulfonic groups are at the edges of the plane [9]. Due to this interesting structure, the growth of MOFs on GO might vary as MOFs interact differently with each type of oxygen functional group. For instance, the favored interaction between MOF–metal sites with carboxylic groups of GO, which are located at the edges of the layers and not on the basal planes, could lead to different crystal growth and arrangements as compared to MOF–epoxy and/or MOF–hydroxyl functional interactions. However, it has been reported that the amount of GO loading also affected the type of oxygen functional group that interacts with the parental crystalline structure [8,32,33,34] and/or the surface of MOFs [35,36]. MOFs usually interacted with the epoxy groups on the GO basal plane when the GO loading was moderate or low in the system, while at a higher GO content, the binding of metal sites to carboxylic groups can become the dominant interaction [8]. The interactions between metal sites of MOFs such as HKUST−1 [Cu–BTC: Cu_3_(BTC)_2_(H_2_O)_3_; BTC = benzene−1,3,5−tricarboxylate] and the oxygen atoms of the epoxy groups have been studied by thermal analysis (DSC). Petit et al. noted the absence of the characteristic peaks belonging to the epoxy groups for their HKUST−1/GO composites, suggesting the growth mechanism of the composite might differ at a higher GO content in their system. Considering carboxylic groups’ higher affinity toward MOF–metal sites than toward epoxy groups, it is reasonable to suggest that the attachment of HKUST−1 on the edges of GO layers might be a preferred mechanism in high GO content [37]. Moreover, at lower GO content, the presence of GO did not disrupt the formation of linkage between metal sites, therefore, retaining the crystalline structure of MOFs while enhancing the surface properties [38]. Daraee et al. found that the enhancement in the specific surface area and the total pore volume of UiO−66 [Zr_6_O_4_(OH)_4_(BDC)_6_] (BDC = 1,4−benzene dicarboxylate) can be attributed to the formation of new pores at the interface between GO and MOFs [35]. Additionally, the microporous channel was adhered to and embedded in the mesoporous channel generating new mesopores to further increase the surface area and total pore volume. However, the incorporation of a higher GO content reduced the size of UiO−66 and caused the MOF crystalline structure to distort. The surface area and total pore volume were also reduced due to the mutilation of the UiO–66 crystalline structure. The same phenomenon was also observed in other types of MOF such as HKUST−1 [8], MIL−101(Cr) [Cr_3_(O)X(BDC)_3_(H_2_O)_2_]·nH_2_O (BDC = benzene−1,4−dicarboxylate, X = (OH or F) [34], MIL−68(In)−NH_2_ [In(OH)(BDC−NH_2_)] (BDC−NH_2_ = 2−amino−1,4−benzene dicarboxylate) [33], and ZIF−8 [Zn(MeIM)_2_] (MeIM = 2−methylimidazole) [36].

The growth direction of MOF varies according to the type of oxygen functional group it interacts with. A study by Asgharnejad et al. demonstrated that the functional groups on GO directed the Ni−MOF [Ni_2_(BDC−NH_2_)_2_(DABCO)∙xDMF∙yH_2_O (Ni−A) BDC−NH_2_ = 2−Aminoterephthalic acid; DABCO = 1,4−diazabicyclo[2.2.2]octane] growth toward the (002) direction according to the proposed mechanism (Figure 2a) [39]. It is evident from their proposed mechanism that Ni−MOFs interacted with the oxygen from the epoxy and hydroxyl functional groups on the basal planes of GO. An XRD analysis of the as-synthesized Ni−MOF/GO composites (Figure 2b) highlighted that the plane of the material had shifted to a lower angle, indicating that Ni−MOFs may be oriented along the [002] direction in the presence of GO [39]. Similarly oriented MOF growth along the [002] plane has also been reported by Liu et al. in the synthesis of a Fe-based MOF composite synergized with graphene aerogels (GAs). After analyzing the interplanar crystal spacing, they found that the preferential growth of MIL−88−Fe was along the (002) direction, thus producing a rod-like MOF on GO [40]. Comparatively, Liu et al. found that nanosized and well-dispersed Cu−BTC also known as HKUST−1 was induced by the incorporation of GO [41]. A hexagonal-shape unit cell along the [111)] direction that belongs to the cubical HKUST−1 crystalline structure was obtained in their HKUST−1/GO composite samples. The epoxy groups on the basal plane of GO not only act as seed sites for HKUST−1 nucleation and crystallization but also prevent the aggregation of nanocrystallites, leading to the formation of nanosized and well-dispersed HKUST−1 crystallites. This finding is in agreement with another study by Petit et al., which reported that the attachment of HKUST−1 blocks to a graphene layer was caused by the interaction between metal sites of HKUST−1 and epoxy groups on GO [13].

Interestingly, the growth directions of MOFs on the GO planes can sometimes also suggest the dominant type of oxygen functional groups that the MOF structure interacts with. As reported by Gu and Zhang, the growth of MOFs toward [110] usually indicates the interactions between MOFs with carboxyl functional groups [42]. As the direction of [220] is parallel to [110], it is safe to assume that the growth of MOFs on GO toward the [220] direction is also due to the interactions between metallic sites of MOFs and the carboxylic functional groups. Further examples were reported by Jahan et al. [31] and Zhou et al. [25]. Both groups found the oriented MOF growth toward [220] could be ascribed to the strong interaction between MOF metallic sites and the carboxylic group functionalized graphene. Kim and Coskun reported the preferential continuous growth of HKUST−1 on the GO surface, with the HKUST−1 structures oriented in the [220] direction (Figure 2c) [43]. The XRD results in Figure 2d show the intensified [220] peak in the as-synthesized HKUST−1/GO composite materials, indicating the crystal structures are oriented toward the [220] direction. However, it is not mentioned which functional group of graphene oxide interacted with HKUST−1. Therefore, it can be concluded that GO oxygen-containing groups are of crucial importance in guiding some MOF growth in a certain preferred direction. However, besides some successful studies mentioned above, it is still challenging to predict the MOF growth and structure orientations due to the richness of MOF compounds and complicated MOF−GO interactions. More systematic studies in this area are highly demanded.

### 2.2. Metal Coordination Environment in Parental MOFs

Another popular group of mechanisms elucidating the growth of MOF/GO composites with oriented MOF structures is the metal coordination environments in the parental MOFs, which could affect the way GO oxygen functional groups coordinate with the metal sites. As known, depending on the types of metals, each MOF has quite a different metal coordination and steric environment [44]. Petit et al. [8] systematically studied the potential growth mechanisms of MOF−5 [Zn_4_O(BDC)_3_, BDC = benzene−1,4−dicarboxylate]/GO, and HKUST−1/GO composites as shown in Figure 3. In the case of MOF−5, a zinc-based MOF, all oxygen atoms as parts of the zinc oxide tetrahedral structure have similar coordination and steric environments in terms of the spatial arrangement (Figure 3a). This can also be applied to zinc atoms in the metal centers. As previously discussed, there is a variety of oxygen functional groups attached to the GO basal planes as well as the edges. Each functional group, therefore, can be coordinated equivalently around the metal centers, forming a regular MOF−5 structure. Their results indicate that the uniform metal coordination environments in the MOF−5 structure played a crucial role in MOF−5 crystallization, regardless of various interactions between the MOF structure and the GO oxygen-containing groups. However, in the case of HKUST−1, which is a copper-based MOF, the coordination of oxygen atoms to the copper site is not all equivalent. This is due to the potential attachment of the GO oxygen groups to copper occurring in either the axial position or the equatorial position (Figure 3b). Therefore, a more disordered HKUST−1 structure compared to MOF−5 was observed in the composite. Figure 3c,d show the proposed structures of HKUST−1/GO and MOF−5/GO composites. The HKUST−1 composite with a more disordered structure shows crystals in irregular shapes (Figure 3e), while MOF−5-based compounds have a more regular arrangement and a layered structure.

As shown in Figure 3f, the MOF−5/GO composite appeared as a layered compound. More interestingly, this layered structure likely corresponded to an alternation between GO layers and MOF−5 blocks. These MOF/GO composites are, therefore, in good agreement with the above-mentioned mechanisms, which were proposed based on the effect of metal coordination in MOFs. Considering the differences between these two composite materials, we believe that the MOF growth in some scenarios was governed not only by MOF interactions with GO oxygen groups but also by the way the metal and oxygen sites coordinated. This effect is mainly affected by the metal coordination environment in parental MOFs. It is also worthwhile mentioning that the ordered/disordered characters of the MOF structures are usually not easily observed and captured under scanning electron microscopes (SEMs). More systematic structure analyses using high-resolution transmission electron microscopes (TEMs) assisted by computational molecular modeling are vitally important and useful to understand the local MOF−GO structure.

It is important to note that regardless of the type of MOFs, their growth on layered GO structures depends on both MOF metal site coordination and their interaction with oxygen-containing groups of GO [13,37,45]. Nevertheless, one or the other may play a more predominant role in guiding MOF growth. However, in very few scenarios when GO functional groups closely interact with the parental MOF metal site, the steric effect might eventually determine the structural growth of the MOF/GO composite [45]. For example, in MOF−5 and HKUST−1 (Figure 4), metal sites were located in parallel or perpendicular planes in the structure. Therefore, the growth of the MOF crystal could proceed as normal due to the cubic nature of the MOF structure. However, in the case of MIL−100, owing to the spherical shape of the pores, the GO layers have multiple ways of coordinating with the MOF units. These various attachments of GO layers could potentially prevent the building of the MIL by blocking the assembly of several cages into a zeolite-like structure (Figure 4, right-hand side). As a result, the resulting MIL−100/GO composite exhibited very poor crystallinity as well as decreased porosity. This finding does not question the existence of interactions between GO and MOF unit cells but rather suggests that the coordination of the two-parent materials (MIL−101 and GO) was not beneficial to MOF nucleation and growth. However, at very low GO content, the detrimental effect was not observed due to the probability of having two or more GO layers attached to MIL cages being significantly reduced.

To date, there are still very limited studies reported focusing on the effect of the parental MOF metal coordination environment on the oriented growth of MOFs on GOs. More studies on different MOF−GO systems are important to consolidate current opinions and also to explore other possible MOF−GO growth mechanisms.

### 2.3. Bottom-Up Strategy for Highly Oriented MOF Formation

Bottom-up strategies have been reported recently for the synthesis of highly oriented ultrathin nanosheet membranes [21,22,23]. Typically, in these strategies, a preformed layer of GO nanosheets was used to control the growth orientation of MOFs underneath or on top of the GO layer. Li et al. reported a bottom-up strategy for the oriented growth of ZIF Zn_2_(benzimidazole)_4_ (Zn_2_(bIm)_4_), using ammonia as a synthetic modulator for the first time and provided fine control over MOF growth [22]. The resulting 2D ZIF nanosheets with a thickness of ~50 nm showed a preferential growth in the [002] direction. These MOF/GO membranes held great promise for large-scale molecular sieving applications. Nevertheless, the orientation of the ZIF−L nanosheets was still unsatisfactory, as some nanosheets showed many defects and cracks, which influenced the structural integrity and eventually the separation performance of the membranes. In addition, the synthesis process was complicated, and membrane reproducibility was a big concern.

At the same time, Hwang’s group [21] employed a similar strategy for the fabrication of a highly oriented MOF membrane on a porous tubular substrate using GO-guided self-conversion of zinc-oxide nanoparticles (ZnO NPs) for the separation of hydrogen/carbon dioxide (H_2_/CO_2_). As shown in Figure 5, this strategy combines two important steps. First, the metal oxide NPs acting as seeds were uniformly deposited on the tubular support surface. Secondly, an ultrathin GO layer on top of the ZnO seeding layer was employed to confine and control the growth orientation of the membrane. Lastly, upon contact with a ligand solution, a continuous membrane was prepared via the self-conversion of the thin layer of metal oxide NPs. Interestingly, the continuous membrane was assembled and intergrown by highly oriented MOF nanosheets. As aforementioned, GO nanosheets have abundant polar oxygen functional groups and an aromatic sp^2^ domain; hence, they can serve as structural nodes and participate in bonding interactions with early MOF structures.

Therefore, the preformed layer of ZnO NPs sandwiched between the GO and surface of the substrate favored the oriented growth in the 2D confinement. The growth mechanism was further explained in Figure 6a. The nucleation of zinc benzimidazole (Zn_2_(bIm)_4_) nanosheets started from the interface between GO and the top of the ZnO NPs layer, due to the C−O/C=O groups on the GO surface tending to react with zinc ion (Zn^2+^) from ZnO NPs and ligand molecules from the synthesis solution. The ZnO NPs then come into contact with GO to generate Zn_2_(bIm)_4_ nanosheets. These early formed nanosheets acted as seeding templates to induce the conversion of the bottom ZnO NPs into ***c***−oriented multi- layered nanosheet Zn_2_(bIm)_4_/GO membrane owing to the restricted 2D confinement. Simultaneously, the ZnO NPs were consumed gradually with the increase in nanosheet membrane thickness. Finally, the ZIF nanosheets were intergrown both vertically and horizontally to form a continuous nanosheet membrane with a highly oriented ZIF−L structure. Without the GO layer, nucleation and self-conversion growth of ZnO NPs take place in various directions, resulting in the formation of a layer of randomly oriented nanosheets (Figure 6b). The GO layer on top of the ZnO NPs played a key role in restricting and guiding the growth direction of the MOF nanosheets, resulting in a highly oriented nanosheet membrane (Figure 6c). It is worthwhile mentioning that the GO layer was possibly interwoven with the ZIF nanosheets to form a ZIF/GO nanosheet membrane interlaced with GO nanosheets. Therefore, GO may remedy some defects possibly generated during the intergrowth of MOF nanosheets, thus leading to a high-quality ***c***−oriented membrane compared with the membrane obtained without a GO layer.

In a more recent study, Nian et al. successfully fabricated a highly oriented Co-based MOF Co_2_(benzimidazole)_4_, nanosheet membrane using the same bottom-up strategy [23]. Graphene oxides were employed to modify the uneven inner surface of the porous tube, forming a smooth and functionalized surface for the subsequent deposition of the Co-based gel and further transformation into a high-quality nanosheet membrane. It has been found that the deposited gel layer played a multifunctional role in oriented membrane growth, i.e., providing a highly active metal source, assisting in the formation of MOF nanosheets, and controlling the membrane thickness. More importantly, the Co-based gel predefined the layered network. This was critical to the formation of oriented Co-based ZIF nanosheet stacking via local restructuring within the gel phase. In the end, the intergrown Co−ZIF nanosheets exhibited a preferential orientation along [002] direction, forming a uniform and continuous nanosheet membrane with an oriented multilayered structure. Similar to the previous study, the authors observed a random growth of ZIFs in the membrane syntheses without GO guiding.

Hence, the above-mentioned bottom-up strategies, which were employed to fabricate oriented MOF membranes, suggested the formation and orientation of the MOF structures could also be affected by GO confinement or guided by preformed GO structures. We therefore anticipate that similar MOF/GO composite structures will be emerging rapidly in the coming years, which could further contribute to the understanding of GO-guided MOF growth.

## 3. Applications of MOF/GO Composites

MOF/GO composite materials have exhibited great synergistic effects as the MOF and GO complement each other, producing suitable materials for target applications. Numerous physical and chemical properties have been found for MOF/GO composites [29,46,47,48,49]. However, the practical application of oriented MOF/GO materials is still at a nascent, yet promising, stage. They have been studied extensively in gas separation, electrochemical, and photocatalysis. In this section, we summarized the promising performance of oriented MOF/GO composite materials in the above-mentioned applications.

### 3.1. Gas Separation

As aforementioned, the highly oriented MOF/GO nanosheet membrane with [002] orientation exhibited excellent gas separation selectivity and permeability. More specifically, Li et al. reported the synthesis of GO-guided ZIF−GO membranes for highly efficient gas separation [21]. The membrane achieved a high hydrogen (H_2_) permeance of 1.4 × 10^−7^ mol m^−2^ s^−1^ Pa^−1^ and an excellent H_2_/CO_2_ separation selectivity of 106. Figure 7a illustrates the long-term thermal stability of the membrane with negligible structural degradation when exposed to a feed gas temperature range of 30 °C to 150 °C for a duration of 400 h. The high H_2_ flux was attributed to the thin activation layer of preferentially intergrown crystals along the [002] direction with low defects and a reduced number of pinholes [21,22,23]. These oriented continuous nanosheet membranes thus showed significantly improved H_2_/CO_2_ selectivity as compared to microporous molecular sieve membranes [50,51,52,53,54,55,56,57]. Petit et al. synthesized different MOF/GO nanocomposites (MIL−100 (Fe)/GO, MOF−5/GO, and HKUST−1/GO) for the adsorption of toxic gases, such as NH_3_, H_2_S, and NO_2_. Extraordinary separation capabilities were obtained for their MOF/GO composites with the HKUST−1/GO composite exhibiting greater adsorption of ammonia and a breakthrough capacity of 231 mg g^−1^ [19,20,37,58,59]. The HKUST−1/GO has various open sites and newly formed adsorption sites located at MOF/GO interfaces, which could significantly enhance the adsorption performance. Figure 7b shows a graphical representation of ammonia adsorption inside HKUST−1/GO structures. There were two types of adsorption sites in the composite structure. Copper metal centers with open metal sites showed uniform intrinsic ammonia adsorption in the microporous MOF structure. However, the pore spaces newly formed at the MOF/GO interface served as the additional adsorption sites for ammonia molecules. Therefore, HKUST−1/GO composites showed superior guest molecule adsorption as compared to HKUST−1 crystals or GO materials alone.

According to the above studies in both membrane-based and adsorption-based gas separations, the oriented structures in MOF/GO composite materials, which could maximize the gas molecule diffusion, adsorption, and size selection, have shown obvious benefits over traditional MOF samples. Given that MOFs had been displayed as excellent materials for carbon dioxide storage, it would be very useful to demonstrate the applicability of MOF/GO composites as a very promising and efficient group of materials in this research area.

### 3.2. Electrochemistry

Based on the large surface area, high porosity, and structure tunability, the MOF-based nanostructured materials have received tremendous attention in the field of electrochemistry. Zhou et al. prepared oriented Cu-based MOF on a carboxylated graphene-modified electrode surface for electrochemical-based hydrogen peroxide detection [25]. Figure 8a shows high density and cross-linked Cu−MOF sheets with a crystal size of 2−5 μm. An XRD analysis of the nanocomposite revealed a strong diffraction peak at the [220] plane (Figure 8b), indicating the preferential growth of the MOF crystals on the graphene surface. The resulting Cu−MOF/graphene composite showed excellent electrochemical response due to the oriented growth of MOF on the COOH-modified graphene surface and therefore demonstrated high electrocatalytic activity toward H_2_O_2_ reduction (Figure 8c). Figure 8d shows the electrochemical impedance spectroscopy (EIS) for the bare electrode, graphene electrode, and Cu−MOF/graphene electrode. The Cu−MOF graphene electrode showed a sharp increase in charge transfer resistance (R_ct_) of about 1.30 ± 0.05 kΩ due to the electron transfer process of the Cu(II)−MOF/Cu(I)−MOF couple. In addition, the as-synthesized Cu−MOF/graphene electrode exhibited a high sensitivity of 0.792 A mol L^−1^ and a low detection limit of 6.7 × 10^−8^, owing to the synergistic catalytic effect on the porous structure, which was attributed to the electron transfer mediating function of the electroactive Cu-based MOFs and highly conductive graphene.

In another study, Jahan et al. [31] reported the use of benzoic acid-functionalized graphene in the synthesis of MOF−5 with preferred orientation in [220]. The benzoic acid carboxylic group in the graphene surface acted as a structural directing agent and promoted the oriented growth of the MOF−5 structure. The intercalation of graphene in the composite structure significantly improved the electrical properties and surface area. Similarly, Hu et al. employed an epitaxial growth strategy in the synthesis of insulating 2D MOFs on pristine graphene surfaces [60]. The epitaxially grown 2D crystal was highly oriented with a large aspect ratio up to ~1500. The strong hybrid structure can efficiently promote MOF electrochemical activity with a small charge transfer resistance of ~200 Ω and a current density of 30 mA cm^−2^ at only 0.53 V. Hence, the high performance of the MOF/GO composite was not only due to the synergetic effects of the integration of MOF/GO but it was also influenced by the high-quality oriented MOF structures.

### 3.3. Photocatalysis

Recently, oriented MOF-based graphene composites have been extensively studied for photocatalytic applications. For example, Heu et al. prepared UiO−66/GO composites for photocatalytic degradation of organic micropollutants [61]. Although the UiO−66 orientation was not mentioned, the XRD pattern in Figure 9a shows the orientation of UiO−66 in the presence of GO was toward [200], as shown by the previously reported data [62]. Figure 9b shows the adsorption capacity and degradation rate (k) of GO, UiO−66, and UiO−66/GO composites. UiO−66/GO composites showed superior performance of adsorption capacity up to 2.5 times higher than pristine UiO−66. As previously discussed, the combination of MOFs with GO increased specific surface area and pore volumes. Therefore, a high GO content caused the deterioration of the crystalline structure and surface properties of the composites. The UiO−66/GO exhibited better photocatalytic performance compared to pristine UiO−66 (Figure 9b). This is because UiO−66/GO has an energy band gap of 3.2 eV (corresponding to a wavelength of 350 nm), which is lower than 3.55 eV (corresponding to the wavelength of 390 nm) for pristine UiO−66. Herein, the presence of GO played a role as an electron acceptor to assist the carrier separation of photogenerated electrons and holes inside the MOF structure. Thus, it enhanced the light adsorption and narrowed the bandgap of composite materials.

In another study, Vu et al. synthesized a Fe−MIL−53/GO nanocomposite for photocatalytic degradation of reactive dyes [63]. Based on the XRD pattern, the orientation of the composite was toward the [200] direction as well [64]. The Fe−MIL−53/GO composite showed better photocatalytic activity with 96% of red RR195 dye degradation in comparison to pristine Fe−MIL−53. The same findings were reported by Wu et al. for MIL−88(Fe)/GO composites [65]. The XRD pattern confirmed the presence of structural orientation in the [100] direction [66]. The MIL−88(Fe)/GO composite showed better photocatalytic efficiency with a fast removal of methylene blue and Rhodamine B dyes in comparison to pristine MIL−88(Fe) and GO. Therefore, we believe that the oriented MOF/GO composites have great potential for many other applications. For example, as highlighted in another review [67], oriented MOF thin films hold great interest in sensing and electrical applications because of their unique optical and electrical properties. It is expected that the highly oriented MOF/GO composite will have better performance due to the presence of oxygen functional groups decorated along the GO planes. The presence of these groups controls the crystallographic growth of MOFs, endowing the composites with enhanced electrical properties to be used in smart devices.

## 4. Conclusions and Perspectives

MOF/GO composites have received tremendous attention due to their synergetic properties, which significantly enhance the performance in multiple applications. Nevertheless, the fundamental understanding of the composite formation has been very limited. Our review highlighted the underlying formation mechanisms that have been reported so far, although more systematic studies in this field are highly demanded. As discussed in this review, GOs played a significant role in guiding MOF nucleation and growth. They can serve as a structural directing agent that influences the growth direction of MOF structures. The metal coordination chemistry of MOFs was also very crucial in affecting the early nucleation and MOF growth orientation on GO basal planes. Different types of MOFs have quite different metal coordination environments, which could significantly influence their interactions with GO layers. The growth orientation of MOFs can also be controlled by constructing GO confinement to guide and direct the MOF crystal/membrane growth.

To date, it is still challenging to determine which mechanism is predominant in the synthesis of a particular MOF/GO composite. However, each MOF/GO composite material may be very likely to have its unique formation pathway and MOF crystallization mechanism. Hence, more fundamental research is highly recommended for consolidating current understanding in this field and, more importantly, exploring new or missing knowledge on GO-guided MOF growth. The fundamental understanding of MOF/GO formation mechanisms could open up many opportunities to tailor their structures and properties to meet specific requirements in many industrially important applications, such as gas separation, adsorption, electrochemistry, and photocatalysis. Despite considerable progress in synthesizing MOF/GO composites, extensive and deeper analyses are also needed to further evaluate their interesting structures. Therefore, we found that future research in the following directions would be of great interest and benefit.

Extensive efforts are needed to further evaluate the nanostructure of MOF/GO composites to fully understand the mechanism behind the structural directing effect of graphene-based nanosheets. The local nanostructure at the graphene/MOF interface should be more focused. The use of multiple advanced spectrometry instruments, for instance, in situ AFM, HR−FESEM, and HR−TEM imaging may provide a good resolution of the crystal structure at the MOF−GO interface [68].

Computational simulation and modeling of MOF structures, MOF−GO interactions, and MOF/GO local structures can provide a more theoretical understanding of the MOF crystallization with GO. Molecular simulation techniques such as molecular dynamic (MD) simulation and density functional theory (DFT) [69] could provide general theories on controlling the growth direction of MOFs.

Research on more applications of oriented direction MOF/GO composite has always been desirable. With the knowledge of controlling the MOF structure and orientation in MOF/GO composites, they are expected to possess improved properties and performance in many applications.

Overall, this review attempted to address the fundamental knowledge of the underlying MOF/GO composite formation mechanisms. We found that there are very limited studies in this emerging field, and the research on MOF/GO composites is still at an early stage. With the rapid development in this field, this family of MOF/GO composite materials is highly likely to become a hot topic among researchers and open up even more promising applications in the near future.

## Figures and Tables

**Figure 1 membranes-12-00501-f001:**
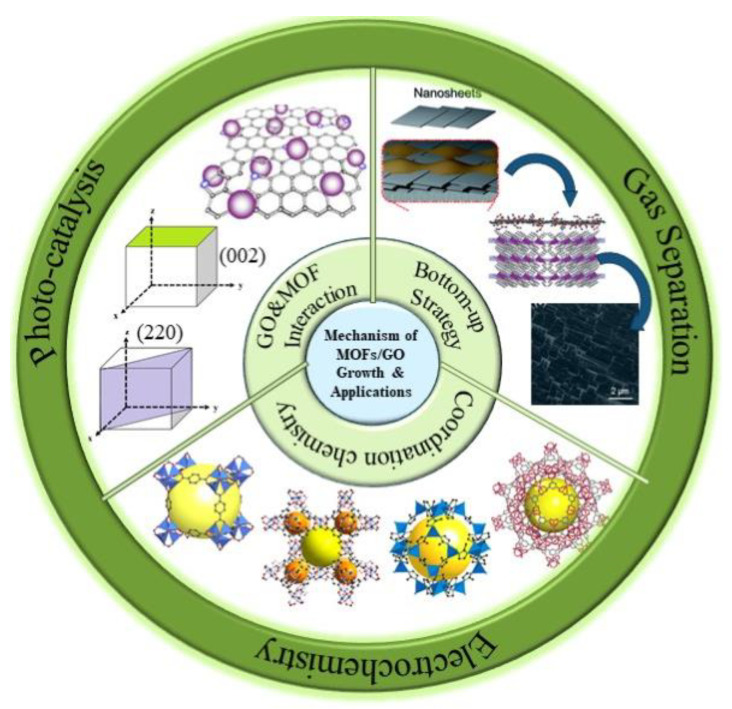
Schematic diagram of factors affecting the growth of metal-organic framework (MOF) in the presence of graphene oxide (GO) and the applications of oriented MOF/GO nanocomposites.

**Figure 2 membranes-12-00501-f002:**
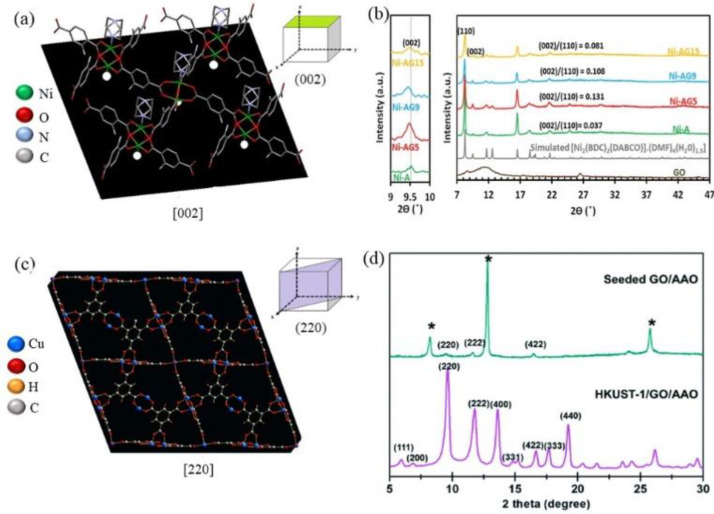
Structural analysis for the influence of GO film on the growth of (**a**) Ni–MOF, (**b**) X-ray Power Diffraction (XRD) patterns of simulated MOFs and composites, (**c**) HKUST-1, and (**d**) XRD data during the consecutive seeding and hydrothermal reactions. Reproduced with permission from [39,43]. Copyright 2017 and 2016, Elsevier and Royal Society of Chemistry.

**Figure 3 membranes-12-00501-f003:**
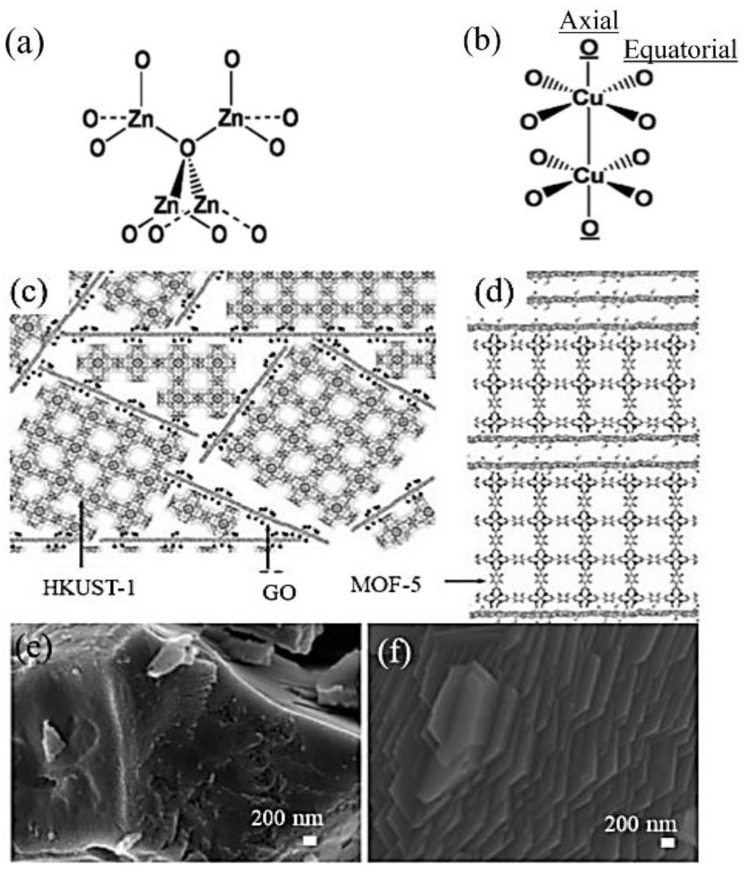
Oxygen coordination on sites available in (**a**) MOF−5 and (**b**) HKUST−1. Envisioned structures for the hybrid materials based on (**c**) HKUST−1 and (**d**) MOF−5; Scanning Electron Microscopy (SEM) images of (**e**) HKUST−1/GO and (**f**) MOF−5/GO. Reproduced with permission from [8]. Copyright 2011, Springer.

**Figure 4 membranes-12-00501-f004:**
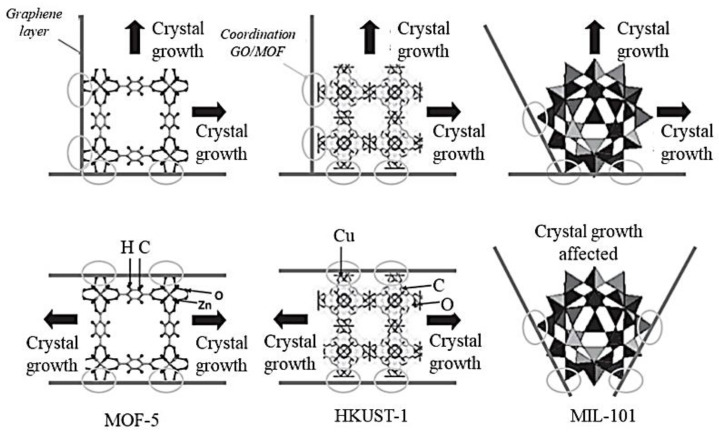
Schematic comparison of the coordination between GO carbon layers to the MOF units for different types of MOF networks: MOF−5, HKUTS−1, and MIL−100. Reproduced with permission from [45]. Copyright 2011, Wiley.

**Figure 5 membranes-12-00501-f005:**
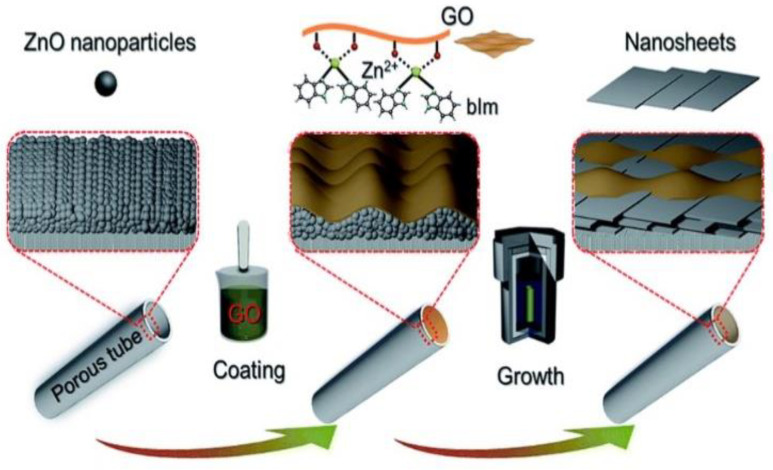
Scheme depicting the preparation procedure of highly oriented Zn_2_(bIm)_4_ nanosheet membranes by ZnO self-conversion growth in a GO-confined space. Reproduced with permission from [21]. Copyright 2018, Royal Society of Chemistry.

**Figure 6 membranes-12-00501-f006:**
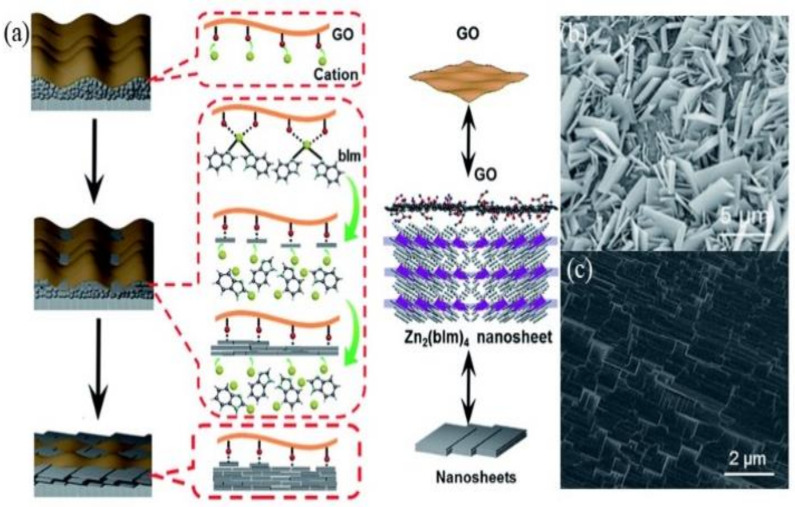
(**a**) Formation mechanism of a highly oriented nanosheet membrane with the aid of GO on porous alumina substrate, SEM images, (**b**) self-conversion of ZnO NPs without GO layer, and (**c**) self-conversion of ZnO NP–GO layer. Reproduced with permission from [21]. Copyright 2018 and 2010, Royal Society of chemistry.

**Figure 7 membranes-12-00501-f007:**
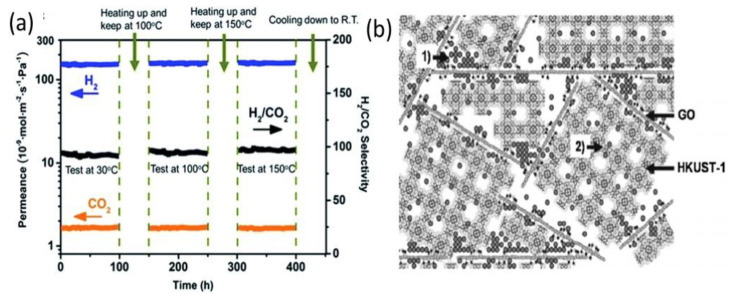
(**a**) Long-term operating stability of the nanosheet membrane in range temperature 30 to 150 °C and (**b**) visualization of ammonia adsorption site in the composite: (1) physisorption at the interface between graphene layers and (2) binding to the copper centers. Ammonia molecules are represented by grey circles. Reproduced with permission from [19,21]. Copyright 2018, Elsevier and Royal Society of Chemistry.

**Figure 8 membranes-12-00501-f008:**
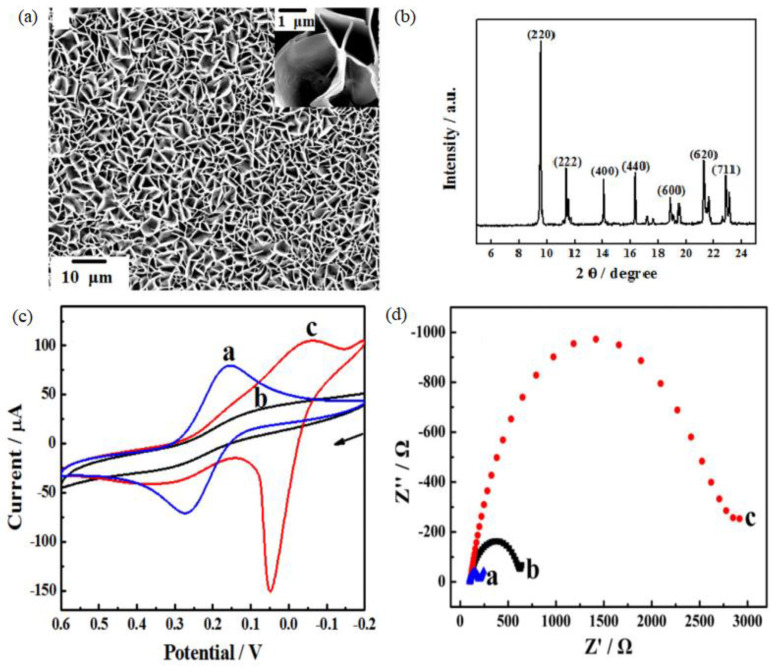
(**a**) Low magnification SEM images, insert high-magnification SEM image of Cu−MOF/graphene electrode, (**b**) XRD pattern of Cu−MOF/graphene-modified bare glassy carbon (GC) electrode, (**c**) Cyclic Voltammetry (CV) of 1.0 × 10^−3^ mol L^−1^ K_3_[Fe(CN)_6_], and (**d**) Nyquist plots of EIS in 5.0 × 10^−3^ mol L^−1^ Fe(CN)_6_^3−/4−^ containing 0.1 mol L^−1^ at different electrodes (a. bare, b. graphene-modified, and c. Cu−MOF/graphene-modified GC electrodes). Reproduced with permission from [25]. Copyright 2018, Elsevier.

**Figure 9 membranes-12-00501-f009:**
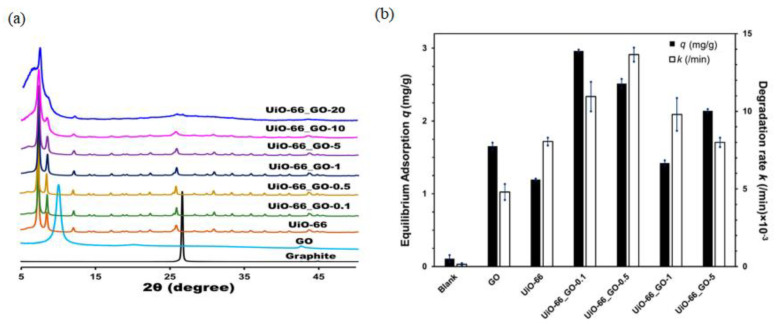
(**a**) XRD patterns of UiO-66/GO composites and (**b**) adsorption capacity and photocatalytic rate (k) of GO, UiO-66, and UiO-66/GO composites. Reproduced with permission from [61]. Copyright 2020, MDPI.

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
