# Peer review of "The Growth of Metal–Organic Frameworks in the Presence of Graphene Oxide: A Mini Review"

_membranes, 2022, doi:10.3390/membranes12050501_

Round 1

Reviewer 1 Report

After reading the manuscript with interest, I do believe this review, may become of interest for readers of Membranes and I do support its publication.

However, I have revisions that I think the authors should addressed before the paper can be accepted

1) The MOF Definition “ Metal-organic frameworks (MOF) are crystalline compounds consisting of metal ions/clusters and secondary building units, also known as organic ligands”  is confusing because  secondary building units (SBUs) can refer to organic ligands but also to inorganic part. SBUs are not exactly a synonym for organic ligands.

In the following reviews the SBUs are well explained

Nature 423, 705–714 (2003).

Chem. Soc. Rev. 2014, 43, 5561–5593

Chem. Soc. Rev., 2009, 38, 1257-1283

2) The authors describe several MOFs. In order to clarify the reading comprehension, the same MOFs should be named in the same way. For example Cu-BTC is the same as HKUST-1 (it would be better to name it always in the same way)

Moreover, it would be interesting to describe the complete formula for each MOF to avoid confusion.

Example:

Cu-BTC: [Cu3(BTC)2(H2O)3; BTC = benzene-1,3,5-tricarboxylate]

MOF-5: [Zn4O(BDC)3], BDC= benzene-1,4-dicarboxylate]

Reviewer 2 Report

In this review, the authors describes the influence of GO on the growth of MOFs and the potential applications of the obtained composites. From a general point of view, results are clearly presented and the topic is of interest. The interaction between GO and MOFs active sites are well discussed. The following comments should be considered by the authors :

  • there are some repetitions in the manuscript that should be carefully checked by the authors.
  • although the manuscript is focused on membrane applications, I suggest to the authors to cite the following papers: J. Environ. Chem. Eng. 2020, 8, 104351; ACS Omega 2017, 2, 4946-4954.
  • the influence of the presence of GO on the size of MOFs crystals and on their specific surface area should be better discussed.
  • paragraph 3.3 : 1) specify the energy bandgap of MIL-88B-Fe, 2) the choice of the MIL-88B-Fe/C3N4 is debatable as no GO is present in the heterostructured photocatalyst (only BE g-C3N4), which is not in accordance with the title of the manuscript.
  • the manuscript contains a few typing errors that can be corrected at the proof stage.

Reviewer 3 Report

The authors reviewed the growth of MOFs into Graphene Oxide that Influences the Growth of Metal-Organic  Frameworks. The submission requires major revision before considering the following points:-

  1. The title should be revised. The current title is irrelevant since the review cannot answer this question precisely.
  2. The authors have to focuss on MOF/GO membrane and their applications for the journal readers.
  3. The authors have to create some new Figures summarizing some points. All the current Figures are reprinted from literature.
  4. The introduction should be rewritten to be broad. The first paragraph for MOFs should be updated with recent references for the topic, including these References; https://doi.org/10.1016/j.ccr.2021.214263; https://doi.org/10.1016/j.ccr.2021.214263; https://doi.org/10.1016/j.chemosphere.2022.134516; https://doi.org/10.1016/j.ccr.2021.214392
  5. The references should be updated.
  6. The language should be revised, and typos should be corrected.

Round 2

Reviewer 2 Report

All corrections were made by the authors. The manuscript can be accepted by Membranes.

Reviewer 3 Report

The authors addressed most of the comments and the revised version can be accepted.